# PyFuRNAce: an integrated design engine for RNA origami

L. Monari [1,2], I. Braun [2,3,4], W. Verstraeten [1], E. Poppleton [1,2,5] &
K. Göpfrich [1,2] ✉

Recent developments in medicine and biotechnology have revealed the transformative power of RNA design. To realize the full potential of RNA nanotechnology and RNA origami, user-friendly design tools are needed. Here, we present pyFuRNAce, an open-source, Python-based software package with a graphical user interface that enables the design of complex RNA nanostructures, with particular focus on co-transcriptional RNA origami. PyFuRNAce integrates the entire RNA origami workflow—from motif definition and blueprint design to sequence generation and primer selection—into a single, user-friendly platform. Built around a motif-based assembly paradigm, the software enables users to create and modify custom RNA nanostructures through an intuitive web interface with streamlined design steps and real-time 3D visualization. We use pyFuRNAce to design three distinct RNA nanostructures, including self-assembling RNA filaments, RNA droplets, and the largest co-transcriptional RNA origami to date, consisting of 2501 nucleotides. The structures and their high-yield assembly are validated experimentally with atomic force microscopy and confocal fluorescence imaging. By consolidating multiple design stages into a unified environment, pyFuRNAce broadens the scope and reduces the barrier of entry for RNA nanotechnology, accelerating the development of functional RNA origami structures for applications in medicine, biotechnology, and synthetic biology.

RNA is widely known as an intermediary between two seemingly more interesting classes of molecules: DNA, for information storage, and proteins, the functional building blocks of a cell. However, the last fifty years of molecular biology have been characterized by an increasing appreciation for the diverse roles that RNA plays in cellular function. Central to its diversity is the ability of RNA to fold into complex secondary, tertiary and quaternary structures—like proteins. Scientists, naturally, asked whether it is possible to design RNA sequences with defined functions and structures, a question that led to the development of techniques like SELEX (Systematic Evolution of Ligands by Exponential Enrichment)[1–3] and the rational design of RNA-based nanostructures[4–6]. The field which emerged from this question, RNA nanotechnology, is still in its infancy compared to protein design and DNA nanotechnology. This is not only due to the historical lack of interest, but also the hard-to-predict dynamic and flexible nature of RNA, and the fact that RNA is built from just four chemically similar nucleotides, which interact in intricate ways owing to additional base pairing possibilities with the 2′ OH group on the ribose—unlike proteins, whose diverse amino acids form a wide range of predictable interactions like hydrophobic cores and salt bridges to guide folding[7] and DNA which has a more constrained and predicable folding landscape. Additionally, the highly charged nature of RNA enhances the effects of buffer conditions on its folding. The number of experimentally characterized RNA structures is a fraction of what is available

[1]Center for Molecular Biology of Heidelberg University (ZMBH), Heidelberg, Germany. [2]Max Planck Institute for Medical Research, Heidelberg, Germany. [3]Max Planck School Matter to Life, Heidelberg, Germany. [4]Max Planck Institute for Dynamics and Self-Organization, Göttingen, Germany. [5]Max Planck Institute for Polymer Research, Mainz, Germany. ✉e-mail: k.goepfrich@zmbh.uni-heidelberg.de

for proteins (200,000 protein structures vs. less than 2,000 RNA structures), making machine learning approaches more difficult[7]. Yet recent years have revealed promising and wide-ranging applications of RNA nanotechnology in biotechnology, synthetic biology, material science and medicine, where aptamer-based drugs are reaching the clinical trial phase[8,9]. Early advancements in RNA nanotechnology demonstrated the structural capabilities of RNA through rational design of discrete and periodic 2D and 3D shapes, such as sheets[10], nanotubes[11], 2D crystals[5], 3D crystals[12], and polyhedra[13]. To accomplish these feats of rational design, the field has developed three RNA design paradigms: RNA architectonics (tectoRNA)[4,14], co-transcriptional RNA origami[10,15], and paranemic cohesion (PX) RNA origami[16], each with their advantages and disadvantages. In principle, RNA origami structures designed for co-transcriptional folding could also be assembled through thermal annealing, as the design algorithms aim for thermodynamically stable configurations, as tested by Geary et. al[10]. However, the development of co-transcriptional RNA origami was preferred for potential applications in vivo or in synthetic biology[17]. In parallel, hybrid RNA-DNA nanostructures are also emerging as a complementary design space[18]. RNA nanostructures have also been used in numerous preliminary studies in medicine[19,20], with applications including siRNA delivery[21], immunomodulation[22,23], and protein inhibition[9,24,25].

Despite progress in RNA nanotechnology, it remains underdeveloped compared to the more established field of DNA nanotechnology[26,27]. DNA nanotechnology has seen tremendous advancements, in particular thanks to the DNA origami design paradigm[28], which enabled the production of megadalton structures at high yield. Although both DNA and RNA origami create raster-filled structures with nanoscale addressability, co-transcriptional RNA origami has some unique advantageous features for applications in medicine, biotechnology and synthetic biology. While DNA origami consists mainly of neatly arranged DNA duplexes, RNA significantly enlarge the design space beyond canonical folds thanks to additional base pairings with the 2'OH group on the ribose. Additionally, co-transcriptional RNA origami can be produced in a single biochemical step, without the need for thermal annealing, offering the potential for mass production and genetic encoding in cells. RNA origami further benefits from a large library of aptamers and ribozymes that can be used to directly integrate functional motifs without covalent chemical modification[8].

Since RNA and DNA origami involve similar material costs (e.g., linear DNA fragments) and experimental techniques (e.g., AFM, TEM, Cryo-EM), we believe that the limited development of RNA origami is highly affected by the lack of user-friendly design tools. In contrast, DNA origami has seen the emergence of a variety of design tools, such as CaDNAno[29], Tiamat[30], ENSnano[31], Athena[32], SARSE[33], MagicDNA[34] and vHelix/BSCOR[35,36], which have been crucial in the field's rapid advancement. These tools have enabled diverse applications and even led to the establishment of companies focused on DNA origami, including its use in vaccine trials[37,38].

As far as RNA design software is concerned, the landscape is relatively sparse. Tiamat was used to design large PX RNA origami[16]. However, it would benefit from means to integrate structures beyond the canonical double-helix. RNA architectonics has benefited from tools such as Assemble2[39,40], which aims at structure prediction and refinement rather than nanostructure design, and Nanotiler[41,42], which offers limited motif diversity and complex manual steps. The Sterna algorithm[43] is integrated into the DNAForge[44] design tool and can be used to generate wireframe polyhedra based on a spanning-tree representation. PyDAEDALUS[45] has been used to design DNA-RNA hybrid[18] structures, and in theory could be used to generate multi-stranded RNA origami polyhedra.

Co-transcriptional RNA origami has one tailored design tool: the RNA Origami Automated Design (ROAD) software package[15]. ROAD is a command-line tool that employs Perl scripts to generate RNA origami structures from blueprint files (Trace_analysis and Trace_pattern), build 3D structures from a fixed motif library (RNAbuild), and perform sequence optimization through inverse folding and sequence symmetry minimization (Revolvr). However, ROAD relies on text-based schematic files that require precise character alignment through manual editing, which is time-consuming and error-prone. Furthermore, adding new motifs to ROAD's motif library is non-trivial, making it difficult to take full advantage of the diversity of functional RNA motifs. In particular large RNA origami structures form at unsatisfactory yield (0% for an RNA origami composed of 2360 nucleotides)[15]. We provide a comprehensive comparison of existing design tools in Supplementary Note S1.

To streamline RNA origami design, we introduce pyFuRNAce[46]. PyFuRNAce is a user-friendly, open-source web application which expands on the paradigm introduced by ROAD. It provides an intuitive graphical user interface (GUI), real-time 3D visualization, and a complete interface for the entire RNA origami design pipeline. Moreover, pyFuRNAce integrates a flexible motif library, enabling users to select or create custom motifs to meet specific design needs. We experimentally validate pyFuRNAce with three distinct RNA nanostructures, including RNA filaments, droplets and the largest RNA origami to date consisting of 2501 nucleotides which folds co-transcriptionally with over 80% yield. These features make RNA design accessible to experts and non-experts alike, empowering researchers to explore the vast potential of RNA origami.

## Results

The primary goal of pyFuRNAce is to streamline the RNA origami design workflow—from motif definition, blueprint design and 3D visualization to sequence generation, template preparation and primer selection—within a unified and user-friendly interface. RNA origami design is often iterative; multiple refinement cycles are necessary to optimize the geometry, sequence and folding. PyFuRNAce supports this process by integrating all major design stages into a cohesive environment, replacing the previously fragmented pipeline that relied on multiple tools (e.g. text editors for blueprint creation, RNAbuild and ChimeraX for visualization, Revolvr and RNAfold for sequence generation and folding validation, separate utilities for DNA conversion, and external calculators for primer melting temperature). Additionally, pyFuRNAce integrates a robust Python library that allows users to design RNA origami blueprints through Python scripts or Jupyter notebooks. Users can automate the design process, generate RNA origami sequences, and customize the workflow. This capability enables the creation of complex RNA structures through code, facilitating large-scale design iterations, systematic optimization, and integration with other computational tools, offering advanced users complete control over the design process.

The user interface is organized into four modules: Design, Generate, Convert, and Prepare (Fig. 1a), each corresponding to a distinct stage in the RNA origami design pipeline. In addition to merging the workflow into a single web application, pyFuRNAce introduces several key features that simplify and enhance RNA origami design (Fig. 1b), including:

- Support for multi-strand design in the blueprint design module, enabling users to visualize partial designs or multimeric assemblies.
- Real-time 3D visualization of RNA nanostructures.
- A rich and expandable motif library to keep the tool updated with the latest aptamer discoveries.
- A user-friendly graphical interface (GUI) that lowers the entry barrier to RNA origami design, with the option to extend functionality through an integrated Python scripting interface.
- A library of existing RNA origami structures that users can easily modify according to their specific needs.

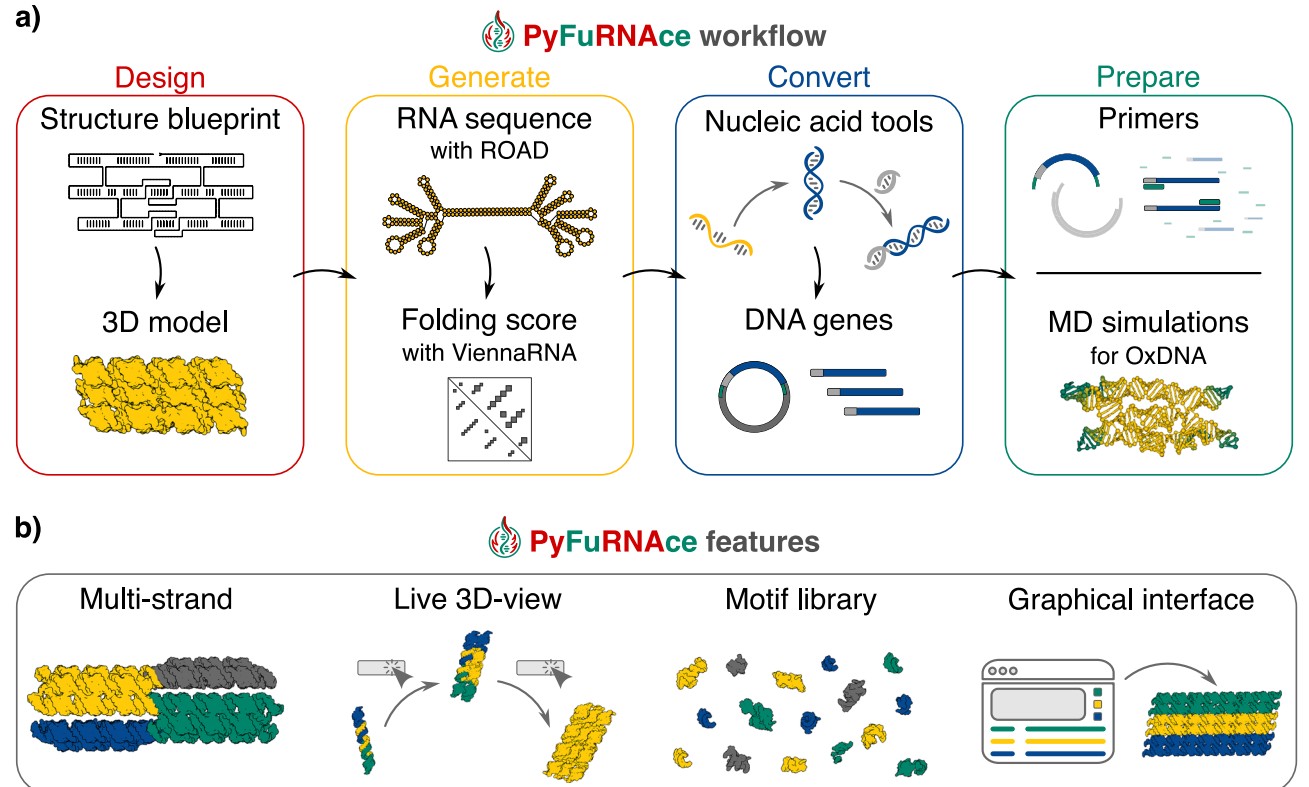

**Fig. 1 | PyFuRNAce workflow and key features. a** The RNA origami computational workflow integrated with pyFuRNAce. The Design module allows for blueprint and 3D structure design; the Generate page produces RNA sequences and structure folding scores; the Convert page analyzes the nucleic acid sequences and outputs a DNA template; the Prepare page helps in the primer design and the preparation of files for oxRNA MD simulations. PyFuRNAce can be accessed at http://pyfurnace.de. **b** New features to streamline RNA origami design introduced with pyFuRNAce.

## Design

The Design module is the platform's central feature. It enables object-oriented motif definition and blueprint generation through a graphical user interface (Fig. 2). At the top of the interface, users can access and insert motifs from the pyFuRNAce library. Below, the RNA origami blueprint and the 3D structure are displayed with interactive components.

Unlike DNA design tools such as CaDNAno[29] or ENSnano[31], which rely on manual strand manipulation, pyFuRNAce adopts a motif-based assembly approach inspired by RNA architectonics and ROAD, extended through object-oriented programming. This motif-based approach is well suited to harness the versatility of RNA folds in RNA design and expands the design space beyond just parallel double helices that make up the bulk of DNA nanostructures. RNA motifs are collections of strands with assigned structures and 3D representations. With motifs as building blocks, an RNA origami is merely a 2D grid in which the motifs are placed. Users can add, remove, or modify motifs while the software dynamically aligns and connects them, assembling the RNA origami blueprint. At the same time, the 3D structure is built by concatenating the motifs' 3D structures, a feature that enables real-time visualization. Importantly, this also enables the design of structures that consist of multiple RNA strands, which was previously not possible with ROAD[15].

All motifs can be modified, changing the sequences, structures, and 2D representations. Among the numerous motifs available in the RNA nanotechnology field, a relatively small number of structural motifs are sufficient to design an RNA origami nanostructure: tetra-loops that cap the end of helices, stems that extend RNA double helices to the desired dimensions, dovetails (stems between crossover junctions) which connect parallel helices at desired angles, and kissing loops, which connect helices within one RNA origami or between multiple origamis (Fig. 3a). In pyFuRNAce, these motifs present extra

features to allow easy customization of properties such as length, sequence, frequency of wobble pairings (i.e. a non-standard base pairing arrangement between nucleotides, such as G-U pairings) or binding energy.

While these motifs are the minimal set to build an RNA origami (Fig. 3b), aptamers or ribozymes are often necessary to design functional structures. PyFuRNAce includes a growing aptamer library for this purpose (Fig. 4a). New aptamers and ribozymes are continuously discovered through experimental techniques such as SELEX[1–3] and recently also AI-based tools[47,48], which promise to tremendously accelerate aptamer discovery in the future. To encourage the expansion of the aptamer library, pyFuRNAce allows users to create custom motifs by defining each strand, their coordinates and 3D structures (Fig. 4b). A dedicated custom motif interface is integrated into the design page to streamline this process (in the motif menu, under the option 'Custom'), with a 'drawing tool' tailored towards motif design.

Recognizing that drawing motifs, either by hand or with drawing tools, can be time-consuming and error-prone, pyFuRNAce includes a Structure Converter tool. This tool converts dot-bracket notation and sequence from FASTA-like files into text blueprints without any need for manual editing. This feature bridges common bioinformatics formats with the pyFuRNAce visual design process, enhancing accessibility and efficiency.

## Generate, Convert, and Prepare

Once an RNA origami structure is complete, the Generate module uses an inverse folding algorithm (Revolvr[15]) to produce an RNA sequence compatible with the target secondary structure (Fig. S3a). The blueprint is converted into sequence constraints and dot-bracket notation compatible with Revolvr and saved as a target file; then Revolvr is executed in the background. It is important to note that Revolvr requires the installation of Perl and does not yet support kissing loop-

## Design

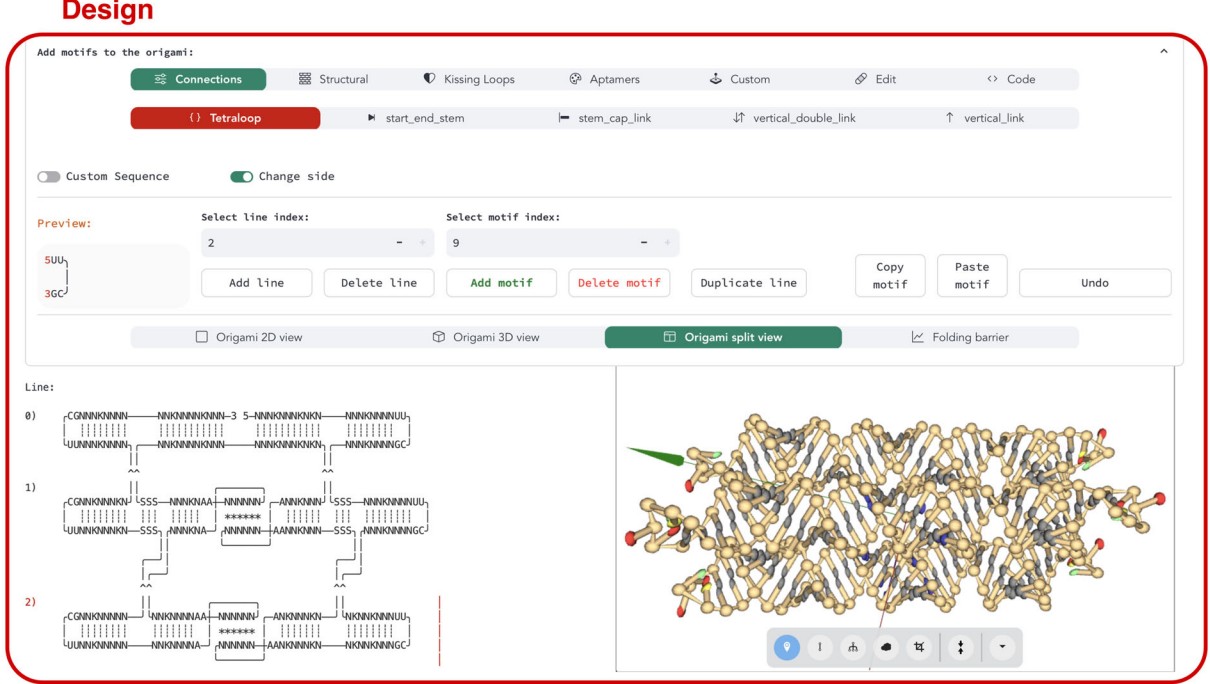

**Fig. 2 | A screenshot of the pyFuRNAce interface (availabe at http://pyfurnace.de).** At the top, the motif menu allows for motif selection, creation, and editing via either a graphical or scripting interface. Below, motif properties and a preview are shown, along with index tabs to position the motif within the origami. At the bottom, a menu controls the display settings for the origami; in this case, the origami blueprint is shown alongside the 3D structure in a split view.

## a) Core motifs

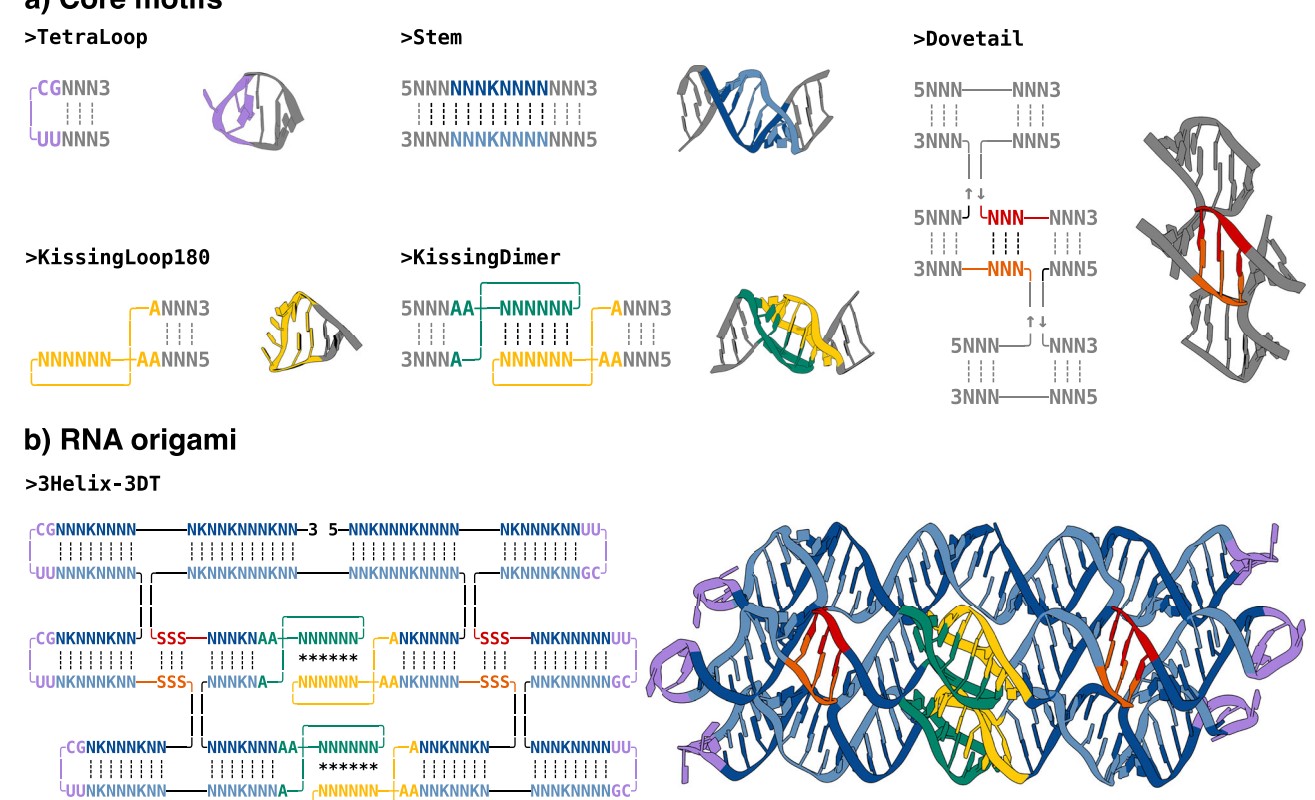

## b) RNA origami

**Fig. 3 | Structural motifs and their integration into RNA origami designs. a** Most common structural motifs for designing an RNA origami nanostructure. For each motif, the name, blueprint, and 3D structure are shown. Nucleotides that are part of the motif are highlighted in colour, while `dummy' motifs are displayed in grey to represent the overall structure, including additional double helices. **b** An RNA origami blueprint with the corresponding 3D structure. The different motifs in the structure are highlighted using matching colours.

**Fig. 4 | The expandable pyFuRNAce motif library. a** The current pyFuRNAce motif library (excluding core motifs), with blueprints and 3D structures. The library includes protein-binding aptamers (yellow), fluorescence light-up aptamers (green), substrate-binding aptamers (blue), and structural motifs (light blue). **b** Custom motif definition: A motif is composed of strands, each defined by strand specific binding energies (see Installation section of the online documentation and Supplementary Note S3). Finally, ViennaRNA folding parameters are provided to evaluate the folding quality, including the energy of the minimum free energy structure, its frequency in the ensemble of possible structures, and ensemble diversity (the characters, start position, start direction, and 3D coordinates. The corresponding blueprint is shown on the right. **c** PyFuRNAce enables the conversion of secondary structure in dot-bracket notation (left) into a corresponding blueprint motif (right), along with an associated sequence.

average weighted distance between all possible secondary structures, Fig. S3b).

The Convert module translates RNA sequences into DNA coding sequences and allows the addition of a sequence as a transcriptional promoter at the 5' end (by default, the T7 promoter is used, Fig. S4). It

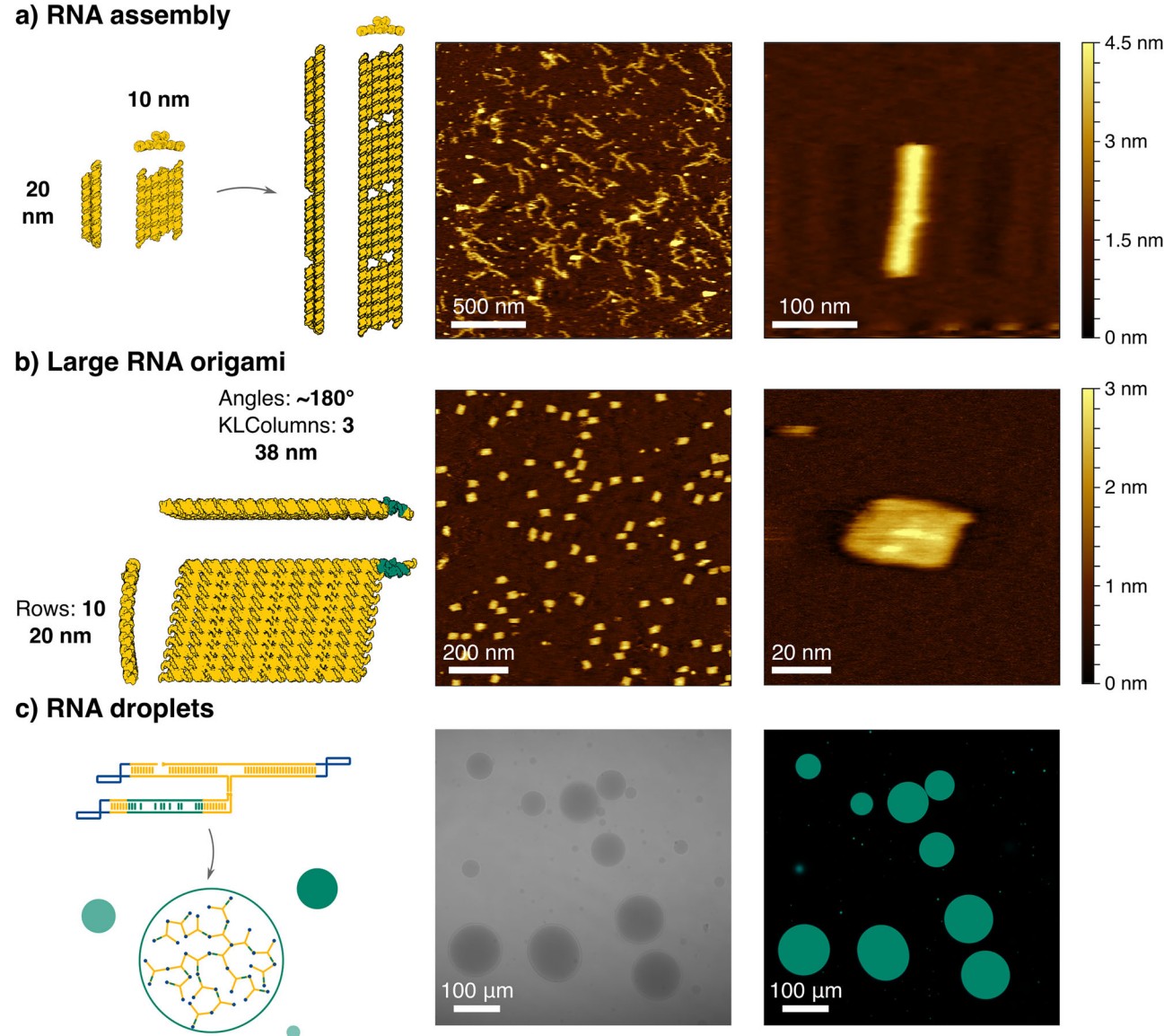

**Fig. 5 | Experimental validation of pyFuRNAce.** Three distinct RNA nanostructures were designed and tested: **a** A two-layer RNA origami tile that polymerizes into filaments. AFM (middle and right image) validates successful co-transcriptional folding according to the designed 3D model (left). **b** A large, rectangular RNA origami structure consisting of 2501 nucleotides with a Broccoli aptamer (highlighted in green). AFM images confirm high-yield co-transcriptional folding. **c** An RNA droplet-forming motif with stems shown in yellow, kissing loops shown in blue and the Broccoli aptamer in green. A bright-field image (middle) and a confocal fluorescence image (right) confirm droplet formation. Each sample was imaged once under AFM or confocal microscopy (n = 1). Confocal source data are provided as a Source Data file.

includes analytical tools for determining GC content, dimer prediction, and subsequence search. The Prepare module supports optional steps such as primer design and oxRNA simulation setup. Primer design is based on melting temperature calculations, with customizable parameters to account for buffer composition and PCR conditions (Fig. S5). Primers can be calculated automatically following the design parameters described in Supplementary Note S2. The oxRNA setup includes default simulation files and parameters for evaluating structural stability, using a protocol that we tailored specifically for co-transcriptional RNA origami (Supplementary Note S3).

### Experimental validation of pyFuRNAce
To validate the capabilities of pyFuRNAce and assess the robustness of its settings, we designed and tested three distinct RNA nanostructures, each leveraging specific features of the software. First, we developed a two-layer RNA origami filament using the default origami generation ('Make simple origami') to control interhelical angles (Fig. 5a; all DNA templates and primer sequences are listed in Supplementary Data 1). The design consists of a base layer (six parallel helices) and a second layer of two central helices, demonstrating 3D structure design enabled by pyFuRNAce's live 3D view. The resulting tile had an approximate width of 20 nm and a height of 10 nm. During the design, the 3D view revealed steric clashes in the base layer due to the rigid dovetail model; however, we hypothesised that the intrinsic flexibility of RNA might resolve such clashes during folding. To design filament assembly, we added terminal 180° kissing loops and used the multi-strand live 3D view in pyFuRNAce to optimise terminal stem lengths. All design scripts, including those for the filament, are provided in Supplementary Note S5. Agarose gel electrophoresis (Figure S6) and atomic force microscopy (AFM; Fig. 5a, Figure S7) confirmed successful co-transcriptional assembly, suggesting that the structural flexibility of RNA can indeed compensate for geometric imperfections in the design model.

Next, we investigated whether the 'Make simple origami' interface could support the scalable design of large RNA origami structures. The largest previously reported co-transcriptional RNA origami, designed with ROAD, consisted of 2360 nucleotides and exhibited poor folding yields, with no defect-free structures observed (reported as 0% correctly folded structures)[15]. Notably, that design employed long dovetail connections (+11 bp), whereas we hypothesised that shorter dovetails might improve co-transcriptional folding efficiency. Using the pyFuRNAce simple origami interface (Supplementary Note S5), we designed a rectangular RNA origami composed of 10 parallel helices connected by three columns of kissing loops, totalling 2501 nucleotides. To enable real-time monitoring of co-transcriptional folding and demonstrate functional addressability, a Broccoli aptamer was integrated into the design. The folding pathway was optimised using pyFuRNAce's default folding features, while sequence optimisation was performed through the Python scripting API. Plate reader assays performed during transcription showed increasing Broccoli fluorescence over time, indicative of co-transcriptional folding (Figure S6), and AFM imaging revealed a high yield of correctly folded structures (Fig. 5b, Figure S8). A raw yield estimation (Figure S9) indicated an unexpectedly high assembly efficiency of 80.29%, supporting the hypothesis that the use of short dovetails and automated folding-path optimisation facilitates the correct folding of large RNA origami structures. To our knowledge, yields exceeding 80% have previously only been reported for RNA origami structures shorter than 400 nucleotides[9,15,17,49].

Finally, we aimed to demonstrate the versatility of pyFuRNAce beyond conventional RNA origami by designing a droplet-forming RNA nanostructure. RNA droplets, or condensates, have been engineered to form phase-separated compartments[49–51]. Previously published RNA droplets are typically based on Y- or X-shaped motifs (i.e. RNA nanostructures with three or four arms) connected via unpaired uracil linkers, with sequence optimisation performed with NUPACK[49,50]. The use of computational tools to prepare the secondary structure, which included the insertion of different aptamers, was not reported. We used pyFuRNAce to design a three-armed Y-shaped motif connected at the centre by unpaired uracils (Supplementary Note S5, 'Droplet-5-in-arm'). Each arm terminated in a palindromic kissing loop, using sequences previously shown to drive droplet formation[49,50]. A Broccoli aptamer was integrated to enable visualisation. Confocal microscopy confirmed the formation of micron-scale RNA droplets (Fig. 5c). As additional validation, we designed a second construct starting the sequence from the Y-core, which also successfully formed droplets ('Droplet-5-in-core', Figure S10). These experiments confirm that pyFuRNAce enables both reliable design and functional expression of diverse RNA architectures. Importantly, it transforms what was previously a fragmented and manual process into a streamlined design pipeline accessible through an intuitive graphical interface, enabling more complex and large designs than could previously be realized. In addition to supporting the creation of novel RNA blueprints, pyFuRNAce facilitates the integration of existing designs from the literature. Many published RNA nanostructures are available only as dot-bracket representations of nucleotide sequences. To bridge this gap, we implemented a tool in the pyFuRNAce interface that converts dot-bracket structures and linear sequences into fully editable RNA origami blueprints. The platform also includes a library of curated templates to facilitate the initiation of new designs, incorporating structures from previous studies[9,15,17,49,50]. When sharing RNA nanostructures in the future (e.g by uploading to Nanobase[52]) we recommend uploading the sequence and dot-bracket as a FASTA file, as well as the PyFuRNAce .py file and the oxDNA 3D structure. This best practice ensures the reusability and composability of RNA designs and will lower the barrier of entry for future entrants to the field.

These features aim to accelerate RNA origami development, improve design reliability, and make advanced design strategies more accessible to a broader community.

## Discussion

RNA origami is a powerful method for designing functional and programmable RNA nanostructures with wide-ranging applications in nanomedicine, synthetic biology, and biosensing. Due to the high stability of folded RNA structures and the possiblility for multiplexing of e.g. aptamers, there is tremendous potential in the use of RNA nanostructures and RNA origami for therapeutic applications. However, the lack of integrated and user-friendly design tools has hindered the broader adoption of RNA origami beyond small expert communities.

PyFuRNAce addresses this gap with several key contributions to the landscape of RNA origami design software:

1. *Integrated graphical interface:* pyFuRNAce offers an intuitive, fully integrated graphical user interface that lowers the entry barrier for new users while providing advanced features for experienced designers. The interface enables real-time updates and seamless transitions across design stages with a few clicks, simplifying a process that previously required multiple disjoint tools. A tutorial video is provided with online documentation and can be found in the Supplementary Information of this manuscript (Video S1).

2. *Custom motifs support:* The rapid discovery of new functional RNA motifs, e.g. through SELEX and enhanced by AI, demands a flexible design environment. PyFuRNAce allows users to define and integrate new custom motifs, including aptamers and ribozymes, integrating the latest structural innovations into RNA origami designs. Additionally, a library of existing RNA nanostructures allows for their straightforward modification towards new research goals, e.g. allowing users to integrate aptamers into an existing design.

3. *Programmability:* While interactive design is essential, some tasks —such as defining scaffold layouts, testing structural variants, or optimizing folding pathways—require extensive user inputs or manual modification. PyFuRNAce's Python interface allows users to build complex structures by scripting, accelerating the design process and enabling high-throughput applications.

Using pyFuRNAce, we were able to generate and experimentally validate a diverse range of RNA nanostructures, including filaments, droplets and the largest RNA origami structure to date consisting of 2501 nucleotides. The large RNA origami rectangle folds at a yield of over 80 %, which has previously only been surpassed for RNA origami nanostructures of less than 400 nucleotides[15]. PyFuRNAce thus paves the way for more reliable and scalable RNA folding.

Despite these advantages, pyFuRNAce has some current limitations. Its blueprint representation is optimized for adjacent motifs, and non-linear connections (e.g., between non-adjacent helices) are not yet straightforward to design. To support cotranscriptional folding, pyFuRNAce includes a simplified folding barrier optimization; however, optimal pseudoknot arrangement still relies on the designer's judgement (Supplementary Note S3). Additionally, the sequence generation engine (Revolvr) lacks support for multi-strand optimization and fine-grained control over individual kissing-loop energies. To address these issues, we are developing a new inverse folding algorithm tailored to RNA origami, to expand pyFuRNAce's sequence design capabilities to match its structural design flexibility. We believe that pyFuRNAce in its current form is a useful tool which will help grow a larger community around RNA origami and that its capabilities will grow with future releases.

In summary, pyFuRNAce provides a comprehensive and accessible platform for RNA origami design, bridging the gap between abstract structural modelling and practical sequence implementation. Integrating design, visualization and sequence optimization into a single environment, pyFuRNAce empowers researchers from diverse fields to easily design RNA nanostructures with speed, accuracy, and creativity.

## Methods

### Software implementation and dependencies

PyFuRNAce was developed and tested using Python 3.12 (minimum supported version: 3.10). The dependencies include Streamlit[53] (v1.43) to build the user interface; and NumPy[54] (v2.2.2) and SciPy[55] (v1.14.1) to handle core functionality for 3D structures. oxDNA Analysis Tools[56] is an optional dependency that supports PDB file reading/writing and preparing designs for oxRNA simulations[57,58]. As of the current version (1.0.0), the pyFuRNAce project consists of approximately 14,700 lines of original Python code (excluding comments and blank lines).

Several external Streamlit components were used to build an interactive and responsive user interface, including st-click-detector[59] for interactive RNA origami blueprint editing, st_oxview[60] for 3D structure visualization using oxView[56,61], and st_forna_component[62] for rendering secondary structures in dot-bracket notation via Forna[63] on the Generate page. PyFuRNAce includes and integrates Revolvr from the ROAD package[15] for RNA sequence generation. The Revolvr algorithm is a core element of pyFuRNAce, since it implements the inverse folding step, generating RNA sequences from target structures. To support different operating systems, we modified the Revolvr script to rely on the ViennaRNA python API[64,65], already required for pyFuRNAce, and not on a locally compiled copy of ViennaRNA.

In the end, the Biopython library[66] is used to implement sequence conversion, melting temperature calculations and primer design. While pyFuRNAce does not include the oxDNA simulation engine[8,67], it generates all required input files for downstream molecular dynamics simulations using the oxRNA force field[57].

The pyFuRNAce package is cross-platform and can be installed on Windows, macOS, and Linux operating systems. Additionally, a pyFuRNAce webapp can be accessed at http://pyfurnace.de. The structure of the pyFuRNAce package, with the integration of external dependencies, is detailed in Figure S1, Figure S2 and Note S3.

### RNA 3D structure generation

All 3D structures in pyFuRNAce are internally represented in the oxDNA format[68], ('new topology' format with 5′ to 3′ directionality). The 3D geometries of common structural elements—such as stems, dovetails, and 180° kissing loops, including their branched and dimeric forms—are computationally generated using OxView parameters[56] for ideal RNA A-form helices. Dovetail crossover geometries are adapted from the average coordinates of crossover points identified in the PDB structure 7QDU[69].

All other 3D motifs used in pyFuRNAce are either extracted from experimentally determined structures available in the Protein Data Bank (PDB IDs are listed in the code documentation) or predicted using RNAComposer[70,71] when no experimental structure is available.

### In vitro transcription (IVT)

DNA sequences for the RNA filament, large origami, and droplets were generated with pyFuRNAce and are provided in Supplementary Data S1 and in the Source Data file.

**Filament and rectangular RNA origami.** DNA templates were synthesised as double-stranded fragments by Twist Bioscience. These fragments were PCR-amplified with the Q5 2X Master-Mix Kit (New England Biolabs (NEB); cat. no. M0492S), following the kit protocol with an annealing temperature of 70 °C. The primer sequences were calculated with pyFuRNAce's automated primer design tool (Supplementary Data 1, Supplementary Note S2 target melting temperature: 69 °C, purchased as custom oligos from Integrated DNA technologies (IDT), purification: standard desalting), and the terminal nucleotides were removed or added to match the melting temperature in the NEB Tm calculator. PCR products were purified using the Qiagen PCR Purification Kit (cat. no. 28106). RNA was transcribed and co-transcriptionally folded in a one-pot reaction containing PCR-purified DNA template (5 ng μL$^{-1}$), rNTPs (2 mM each, Jena Bioscience; cat. no. NU-1014S), and 1X T7 transcription buffer provided with the polymerase. Reactions were initiated by adding 24 U of T7 RNA polymerase (ThermoFisher Scientific; cat. no. EP0111) and incubated at 37 °C for 2 h in a final reaction volume of 40 μL. For transcription of the large rectangular RNA origami (Rect-10H-3X), DFHBI-1T (final concentration: 15 μM, from Sigma-Aldrich; cat. no. SML2697-5MG) and KCl (final concentration: 50 μM) were included in the transcription mix to promote proper folding and fluorescence of the Broccoli aptamer.

**RNA droplets.** RNA droplets were transcribed following an established protocol[51]. In brief, DNA templates were ordered from Integrated DNA Technologies (IDT) as gBlocks. DNA templates were dissolved in nuclease-free water and used directly for IVT. Transcription was carried out in a homemade transcription buffer (100 mM HEPES, 20 mM Mg(CH$_3$COO)$_2$, 10 mM rNTPs (2.5 mM each), 1.5 mM DTT, 1.5 mM spermidine, 66.6 mM sucrose) and a 1:10 volumetric dilution of the HiScribe T7 RNA Polymerase Mix from NEB (cat. no. E2040S) and a final DNA concentration of 2,5 ng/μL. 50 μM of DFHBI-1T (Sigma-Aldrich; cat. no. SML2697-5MG) was used. RNA polymerase was always added to the reaction mix as a last step. After mixing, 40 μL of the mixed sample was added to the microscopy observation chamber and stored at 37 °C for 24 h prior to imaging. Each IVT sample was imaged once (n=1).

### Atomic force microscopy (AFM)

AFM imaging was performed using a NanoWizard 2 high-speed atomic force microscope (Bruker) in AC fast imaging mode, equipped with FASTSCAN-D cantilevers (Bruker Nano Inc., Camarillo, CA, USA; $f = 110$ kHz, $k = 0.25$ N m$^{-1}$). The imaging substrate was mica (diameter: 0.95 cm; Science Services GmbH, Munich, Germany; cat. no. E71856-01), glued into a small petri dish (Corning, diameter: 35 mm; height: 10 mm). Each IVT sample was imaged once ($n = 1$), with about two aliquots (0.5 μL each) of the IVT reaction deposited onto the mica substrate. No purification of the IVT samples was performed prior to imaging.

**Rect-10H-3X.** To image the large 2501 nucleotide Rect-10H-3X RNA origami, freshly cleaved mica was covered with imaging buffer (1× TAE, 20 mM MgCl$_2$). Subsequently, 1 μL of the in vitro transcription (IVT) sample was gently pipetted onto the mica surface.

**Filament.** For imaging the filament-forming RNA origami tiles, 100 μL of sodium buffer (1× TAE, 100 mM NaCl) was deposited onto freshly cleaved mica, followed by the addition of 0.3–1.2 μL of IVT sample. The presence of sodium ions helps prevent aggregation and clumping on the mica surface. To gradually exchange the sodium buffer with imaging buffer, 10 μL was carefully removed from the top of the droplet and replaced with 10 μL of imaging buffer (1× TAE, 20 mM MgCl$_2$), repeating this step until filament formation was observed. For improved imaging of small features or window regions, 1–5 μL of 50 mM NiCl$_2$ was added to the sample to enhance interaction between the RNA origami and the mica surface. All AFM images were pre-processed with the open-access software Gwyddion[72] using standardized Align Rows and Levelling tools and then exported into SVG file format.

### Agarose gel electrophoresis

A 1.75% agarose gel was prepared in 1× TAE buffer and precast with 1 × GelRed (Sigma-Aldrich; cat. no. SCT123). PCR-amplified DNA (100 ng, Lane 2 and 5) and 6 μL of IVT reactions were loaded together with purple loading dye (no SDS; NEB cat. no. B7025S, Lane 3 and 6), alongside a 1 kb+ TriDye™ DNA ladder (NEB, Lane 1; cat. no. N3272S). For DNase treatment, an additional aliquot of the IVT sample was incubated with DNase I-XT (NEB; cat. no. M0570S) at a final

concentration of 0.1U/μL for 30 minutes at 37 °C (Lane 4 and 7). All samples were run in 1X TAE buffer for approximately 60-90 minutes at 65 V and imaged under UV light using a ChemiDoc MP Imaging System (BioRad).

## Plate reader assay

Fresh IVT reactions of the Rect-10H-3X large RNA origami were pipetted in triplicates ($n = 3$) into a black, clear-bottom 384-well plate (Greiner; cat. no. 781906) and incubated at 37 °C in a Tecan Spark plate reader. Fluorescence (from the Broccoli aptamer and DFHBI-1T) was measured every minute for 2 hours using excitation/emission settings of 470/505 nm and the manual gain set to 90%. Control reactions lacking the DNA template were included to account for background fluorescence. A triplicate blank sample containing nuclease-free water was also included. Fluorescence intensity over time was used as an indication for RNA synthesis and proper folding of the Broccoli aptamer.

## Confocal microscopy

For confocal microscopy of the RNA droplets, a microscopy cover slide attached to a 6-channel Ibidi sticky slide VI 0.4 (cat. no. 80601) was passivated with 5% (W/V) polyvinyl alcohol (Sigma-Aldrich; cat. no. 363170) rinsed with nuclease-free water and dried under nitrogen flow. Subsequently, the prepared droplet sample was added to the channels of the Ibidi slide and the channels were immediately sealed using a two-component glue (Picodent; cat. no. 1300 1000) to prevent evaporation. The RNA droplets were visualized using a confocal laser scanning microscope (LSM 900, Carl Zeiss AG). A 20× objective (Plan-Apochromat 20×/0.8 M27, Carl Zeiss AG) was employed for image acquisition and the internal magnification was set to 0.5×. Prior to imaging, the incubation chamber of the confocal microscope was heated to 37 °C for at least 30 minutes. RNA droplets are shown in blue, with the broccoli aptamer labelled with DFHBI-1T, $\lambda_{ex} = 488$ nm.

## Reporting summary

Further information on research design is available in the Nature Portfolio Reporting Summary linked to this article.

## Data availability

All data generated or analysed during this study are included in this published article (and its supplementary information files). Source data are provided with this paper. The raw experimental data generated in this study, including AFM files, confocal microscopy images, agarose gel images, and plate reader measurements, have been deposited in Zenodo under accession code https://doi.org/10.5281/zenodo.17180007. Source data are provided with this paper.

## Code availability

The pyFuRNAce source code has been deposited on Github at the repository: https://github.com/Biophysical-Engineering-Group/pyFuRNAce under a GPL-3.0 license. The documentation, with Python scripts and Jupyter Notebooks examples, is available on Read the Docs at https://pyfurnace.readthedocs.io/en/latest/. The package can be installed via the Python Package Index (PyPI) using the standard Python package installer (pip). The web application is hosted on Streamlit Cloud and is freely accessible at: http://pyfurnace.de. The specific version of the code associated with this publication (release "1.0.0") is archived in Zenodo under accession code 10.5281/zenodo.17177893[46].

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

## Acknowledgements

This work was supported by the ERC Starting Grant ENSYNC (No. 101076997, to K.G.), Deutsche Forschungsgemeinschaft (DFG, German Research Foundation) under CRC 392 and CRC 1638 (to K.G.), a research grant from HFSP (Ref.-No: RGP003/2023, https://doi.org/10.52044/HFSP.RGP0032023.pc.gr.168589, to K.G.), the Alfred Krupp Förderpreis (to K.G.) and state funds approved by the State Parliament of Baden-Wrttemberg for the Innovation Campus Health + Life Science Alliance Heidelberg Mannheim (to E.P. and K.G.). The authors thank the Max Planck Society for access to

computational resources. We thank Dr. Cody Geary and Dominic Kempf for their feedback.

## Author contributions

L.M. implemented the pyFuRNAce software with help of I.B. L.M. and E.P. conceptualized pyFuRNAce. L.M. designed and characterized the RNA origami, W.V. designed and characterized the RNA droplets. L.M., E.P., and K.G. wrote the manuscript. K.G. supervised the research. All authors reviewed and approved the final version of the paper.

## Funding

## Competing interests

The authors declare no competing interests.
