## [Transparent Peer Review file · Nature Communications]

PyFuRNace: An integrated design engine for RNA origami

Corresponding Author: Professor Kerstin Göpfrich

Version 0:

Reviewer comments:

Reviewer #1

(Remarks to the Author)

Monari et al. present an open-source software platform, PyFuRNace, designed to streamline the co-transcriptional design of RNA origami. The tool offers a unified graphical user interface that integrates the entire design pipeline—from structural blueprinting and sequence generation to 3D visualization, primer design, and simulation setup. By employing a modular, motif-based assembly strategy, PyFuRNace supports multi-strand RNA structures, real-time 3D preview, and customizable scripting through a Python interface. The inclusion of an extensible motif library and an intuitive GUI significantly lowers the entry barrier for newcomers to RNA nanotechnology, addressing a clear need in the field. The open-source nature and web-based deployment further enhance accessibility and potential community engagement.

Despite the tool's strong design and potential utility, the manuscript faces a critical limitation that impedes its suitability for publication in Nature Communications: the absence of experimental validation. The study centers solely on computational demonstrations and software functionality. Although the reconstruction of previously reported structures (e.g., nanotube tiles) showcases PyFuRNace's capabilities, these demonstrations fall short of establishing its effectiveness in guiding the de novo design, synthesis, and characterization of novel, complex RNA origami architectures. Ultimately, the utility of a design tool hinges on whether it can reliably produce functional, experimentally verifiable structures.

Moreover, while the platform offers improved usability and workflow integration, it appears to build upon existing frameworks—particularly ROAD—without introducing significant innovations in algorithmic principles or design performance. The claims regarding enhanced design reliability and favorable folding pathways remain theoretical, lacking empirical support. In its current form, the manuscript reads more as a software methods paper than a research article advancing the frontiers of RNA nanotechnology. Unless supported by robust experimental data demonstrating capabilities beyond existing tools, the work would be more appropriately positioned in journals focused on bioinformatics tools or synthetic biology platforms, rather than in Nature Communications.

For this manuscript, several issues require further clarification:

1. Experimental verification must be included: This is the most critical point for whether this paper can be accepted. The author should use PyFuRNace to design one or more RNA origami with novel topology or function from scratch. The design should highlight the unique advantages of PyFuRNace, such as taking advantage of its custom motif function or multi-chain design capability. Subsequently, the author needs to: Synthesize the RNA origami by in vitro transcription. Preliminary verification of its folding correctness and yield by gel electrophoresis (Native PAGE). Image the morphology of the nanostructure by atomic force microscopy (AFM) or transmission electron microscopy (TEM) and compare it directly with the 3D model generated by PyFuRNace. If the design contains functional elements (such as aptamers), corresponding functional experiments (such as binding experiments) should be performed to verify its activity. Only by providing these experimental data can the core argument of the paper - that PyFuRNace is a reliable and accelerated engine for RNA origami development - be convincing.
2. Claims on reliability should be more rigorous: The authors should adjust their wording without experimental data to support it. PyFuRNace undoubtedly improves the efficiency and convenience of design, but it is premature to claim that it improves reliability. The authors should clearly point out that the reliability of the tool currently relies on the intrinsic performance of its integrated third-party algorithms (such as RevolvR and ViennaRNA), while PyFuRNace assists designers to make more reliable decisions by optimizing the folding path checker and providing real-time visual feedback.
3. Add detailed comparison with other tools: Although the paper mentions tools such as ROAD, Tiamat, and Assemble2, the comparison is too brief. It is recommended to make a detailed functional comparison between PyFuRNace and existing key tools in the form of a table, from multiple dimensions such as supported design paradigms (co-transcription/TectoRNA, etc.), user interface (GUI/command line), whether it supports multi-chain, 3D visualization, custom motifs, script programming, simulation integration, etc. In particular, if PyFuRNace has advantages in specific performance or functions that other design

methods cannot achieve at all.

4. Provide a more thorough discussion of the limitations of the dependent tools: The authors acknowledge some of the limitations of the RevolvR sequence generation engine in the Discussion section, such as the lack of fine control over multi-strand optimization and kissing loop energies. This is worth acknowledging. To provide readers with a more comprehensive understanding, it is recommended that the authors explain earlier and more clearly in the main text how these limitations directly affect the design capabilities of the current version of PyFuRNACE, and explain how users should circumvent these issues during the design process.

5. Provide a more thorough discussion of the limitations of the dependent tools: The authors acknowledge some of the limitations of the RevolvR sequence generation engine in the Discussion section, such as the lack of fine control over multi-strand optimization and kissing loop energies. This is worth acknowledging. To provide readers with a more comprehensive understanding, it is recommended that the authors explain earlier and more clearly in the main text how these limitations directly affect the design capabilities of the current version of PyFuRNACE, and explain how users should circumvent these issues during the design process.

In summary, PyFuRNACE demonstrates strong potential to become a widely adopted and impactful tool in the field of RNA origami, facilitating the design process and accelerating progress in RNA nanotechnology. However, the current manuscript lacks experimental validation of the reliability and structural fidelity of the designs generated by the software. Moreover, the innovation and performance claims made in the paper remain unsubstantiated and are likely to be met with skepticism in the absence of empirical evidence. Therefore, in its present form, I do not believe the manuscript meets the publication standards of Nature Communications.

That said, if the authors can provide experimental demonstrations showing that PyFuRNACE enables the successful design and realization of RNA origami structures that are difficult or infeasible to construct using existing tools—such as ROAD—and further establish its general applicability across diverse design challenges, the work would have the potential to make a substantive contribution to the field. Such evidence would significantly strengthen the case for publication in a high-impact journal.

(Remarks on code availability)

Reviewer #2

(Remarks to the Author)

The article presents an integrated workbench-type design tool pyFuRNACE for supporting the design of RNA nanostructures in the "RNA origami" paradigm introduced by C. Geary, P. Rothmund and E. Andersen in Science 2014 and successfully pursued in a followup paper in Nature Chemistry 2021. The pyFuRNACE software integrates a textfile-based core toolkit ROAD (Nat Chem 2021), developed mostly by C. Geary, together with several visualization and analysis tools into a convenient and easy to use design platform with a nice GUI, a clear workflow, and no need to move and convert intermediate files between a mishmash of disjointed special-purpose tools.

In a broad classification of DNA/RNA nanotechnology design tools, pyFuRNACE is *motif-based*, that is, the target structures are developed bottom-up by combining elementary design motifs (helices, crossovers, pseudoknots, aptamers,...) in a well-defined manner. (PyFuRNACE also provides the useful possibility of extending the family of design motifs according to the user's needs.) In this pyFuRNACE falls between strand-based tools, such as caDNA and ENSnano, and top-down model-driven tools such as vHelix and DAEDALUS. (All of the latter of course being DNA, not RNA design tools.) The tradeoff in going from strand-based to motif-based to model-based tools is that as the convenience of use increases, the level of detailed control decreases. In the particular case of pyFuRNACE, while it is convenient and effective for pursuing designs in the RNA origami framework, it is also very specific to this particular way of doing things.

Thus, for people working in the motif-based RNA origami framework, which seems to be growing into the mainstream approach to RNA nanostructure design, pyFuRNACE will be a welcome work-saver; but outside of this framework it will naturally be less relevant.

In addition to this general assessment, I have a number of comments and questions to the authors:

Abstract:

You might want to make it more clear that pyFuRNACE is specifically targeted to support RNA origami types of designs. (Cf. "pyFuRNACE [...] enables the design of complex RNA nanostructures", "pyFuRNACE reduces the entry barrier for RNA nanotechnology".)

Introduction:

P. 2, l. 6: Provide some background to the statement "Scientists, naturally, asked whether we would be able to design RNA sequences with defined structures and functions." In general or for some specific purpose? Citations?

P. 2, l. -6 ff.: "Three RNA design paradigms: RNA architectonics, co-transcriptional RNA origami, paranemic cohesion RNA origami". Would also annealed RNA origami be relevant? What about hybrid RNA-DNA designs, which seem to be a growing trend, perhaps even more so than pure RNA origami?

P. 3, I. 4: "DNA origami design... arbitrary(?) structures..." -> "very general structures"

P. 3, II. 4-5: "DNA origami... Although the names are similar, co-transcriptional RNA origami has some unique features". What is the actual structural or functional similarity underlying the similar names?

P. 3, II. 14-15: "The limited development of RNA origami can largely be attributed to one key factor: the lack of user-friendly design tools." Is this really *the* limiting factor? What about complexity of experiments or challenging/costly materials?

P. 3, II. 16-17: In parallel to this list of tools, in particular when listing Athena from the Bathe group, it would be appropriate to mention also vHelix/BSCOR from the Högberg group, which is fundamentally similar but earlier, and better known & more used than several of the others.

P. 3, II. 25: "As far as RNA design software is concerned, the landscape is relatively sparse. Tiamat... Assemble2..." What about pyDAEDALUS from the Bathe group or DNAForge from the Orponen group?

Materials and methods:

General question: Since pyFuRNace integrates so many existing tools (ROAD most significantly, but also ViennaRNA, Forna, oxView,...), how much new code has actually been contributed by the pyFuRNace team? Also, all the other integrated tools are multipurpose, but ROAD seems to be really the engine at the core of pyFuRNace: is it now appropriately credited?

More generally, it would be nice to have a clear listing or diagram of which external libraries have been integrated, to which parts of pyFuRNace, in which way (source, binary, API,...), and what the reuse licenses of each of these libraries are. I was looking for something like this in the SI but couldn't find the information.

Results/Generate, Convert, and Prepare:

General question: The article contains now no experimental validation of the designs created with pyFuRNace. Would such validation be needed, and if not then please present an argument in the paper why the designs should be trusted, and/or if there are some particular cases where caution should be exercised.

P. 11, II. 16-17: "It is important to note that RevolvR [...] does not yet support kissing loop-specific binding energies." Please explain what the issue is here, and provide some pointer(s) to the literature where this issue is discussed.

Programming versatile RNA origami:

P. 12, II. -6 ff.: Is this the script in SI Note S3? If so, then please insert here a pointer to it, and maybe some additional commentary in the Note itself.

P. 13, II. 3-4: "If a pseudoknots pairs before a stem is completely formed, threading of the strand through the duplex is kinetically unfavored and thus is unlikely to form co-transcriptionally." Please explain more clearly what the issue is here, and provide some pointer(s) to the literature where this issue is discussed. Is the proposed optimization routine guaranteed to resolve this issue?

Discussion:

P. 14, II. 4-5: "[T]he lack of integrated and user-friendly design tools has hindered the broader adoption of RNA origami." What about e.g. challenges in the laboratory processes?

P. 15, II. 11-13: "Additionally, the sequence generation engine (RevolvR) lacks support for multi-strand optimization and fine-grained control over individual kissing-loop energies. To address these issues, we are developing a new inverse folding algorithm [...]" Cf. comment to p. 11, II. 16-17: Please explain what is/are the actual technical issues here, and how are they addressed now that the new inverse folding algorithm is not yet available?

References:

The capitalization of article and journal titles is quite inconsistent across the reference list.

Ref. [29]: The bibliographic information here is incomplete.

(Remarks on code availability)

I conducted a cursory review of the code, which seemed very cleanly and professionally done; but I did not look too much into the details. In addition to testing the server-hosted version I also performed a local installation on my Mac following the instructions given in the documentation. Everything went smoothly except for me needing to upgrade my Python version to 10.3 as required by the tool. -- As noted in my review, one thing I felt was missing in the documentation was a clear map of the dependencies on external libraries (incl. current version requirement, availability, license). A small general concern with all tools relying heavily on external libraries is that they may and will break down eventually when the libraries are updated,

so require committed maintenance.

Reviewer #3

(Remarks to the Author)

The manuscript by Morani et al. entitled "PyFuRNace: An integrated design engine for RNA origami" describes a new user-friendly software platform that facilitates the design of RNA origami nanostructures. Apart from making the design workflow for RNA origami more easily accessible, the most noteworthy new features are the ability to design multi-stranded structures, a live 3D view, a large and extendable motif library, and the ability to automate design by scripting. The work is significant for the field since it (compared to earlier published scripts for RNA origami design) lowers the barrier for researchers to make their own advanced designs. The main result of the manuscript is a well-functioning platform and the design examples stay within realm of already experimentally verified RNA origami structures and multimer assemblies. The PyFuRNace software package is professionally done, extendable, and very well documented. The accessibility of the software would benefit from a few video tutorials that show how to design concrete examples e.g. the nanotube assembly and scripting of larger sheets and bundles. The work represents an important contribution and will be highly useful for researchers in several areas of RNA research, and I therefore highly recommend its publication.

Ebbe S. Andersen

(Remarks on code availability)

The code for PyFuRNace is maintained on Github and is well documented. Looking through the code it is well annotated, which will allow other researchers to easily extend the code.

Version 1:

Reviewer comments:

Reviewer #1

(Remarks to the Author)

The authors have addressed all my questions. I recommend the manuscript to publish in NC.

(Remarks on code availability)

Reviewer #2

(Remarks to the Author)

The authors have addressed all the main questions in my earlier review most commendably, and the manuscript has in particular been much strengthened by the now included experimental validation of novel RNA structures designed using the pyFuRNace tool. Also the clarity, precision and coverage of the writing has been improved quite a bit in several places. I hope the authors still have the opportunity to fix the few remaining small typos in the main manuscript and in the captions of the SI figures, as well as check the list of references once more for correctness and formatting (e.g. ref. [65]).

One question that the authors could have addressed more explicitly in their revised version of the manuscript is the kinetic folding barrier mentioned on p. 13, ll. 3-4 of the original submission. This is discussed in the authors' rebuttal letter, but the issue seems to have been omitted from the revised manuscript. (The rebuttal letter indicates that "in the main text, we included the reference to the original work where the folding barrier is defined", but I couldn't identify this reference. The terms 'pseudoknot' or 'folding barrier' also no longer appear in the revised manuscript.) While this issue does not seem to have been a concern in the experimental work now presented in the paper, maybe the authors could still add some notes on this potential concern in the Discussion section of the final publication?

Other than that, I am quite favorably impressed by the quality of the article. I am convinced that pyFuRNace will become a widely used tool in designing RNA origami nanostructures and will contribute significantly to the advancement of the area of RNA nanotechnology.

(Remarks on code availability)

I reviewed the code and tested the tool in my review of the original submission, and was convinced of its engineering quality.

Point-to-point response

Reviewer #1

“Monari et al. present an open-source software platform, PyFuRNace, designed to streamline the co-transcriptional design of RNA origami. The tool offers a unified graphical user interface that integrates the entire design pipeline—from structural blueprinting and sequence generation to 3D visualization, primer design, and simulation setup. By employing a modular, motif-based assembly strategy, PyFuRNace supports multi-strand RNA structures, real-time 3D preview, and customizable scripting through a Python interface. The inclusion of an extensible motif library and an intuitive GUI significantly lowers the entry barrier for newcomers to RNA nanotechnology, addressing a clear need in the field. The open-source nature and web-based deployment further enhance accessibility and potential community engagement. Despite the tool’s strong design and potential utility, the manuscript faces a critical limitation that impedes its suitability for publication in Nature Communications: the absence of experimental validation. The study centers solely on computational demonstrations and software functionality. Although the reconstruction of previously reported structures (e.g., nanotube tiles) showcases PyFuRNace’s capabilities, these demonstrations fall short of establishing its effectiveness in guiding the de novo design, synthesis, and characterization of novel, complex RNA origami architectures. Ultimately, the utility of a design tool hinges on whether it can reliably produce functional, experimentally verifiable structures. Moreover, while the platform offers improved usability and workflow integration, it appears to build upon existing frameworks—particularly ROAD—without introducing significant innovations in algorithmic principles or design performance. The claims regarding enhanced design reliability and favorable folding pathways remain theoretical, lacking empirical support. In its current form, the manuscript reads more as a software methods paper than a research article advancing the frontiers of RNA nanotechnology. Unless supported by robust experimental data demonstrating capabilities beyond existing tools, the work would be more appropriately positioned in journals focused on bioinformatics tools or synthetic biology platforms, rather than in Nature Communications.”

We thank the reviewer for their positive feedback on the software tool itself - community engagement is what we want to achieve through the development of a user-friendly open-source tool. We agree that experimental validation is essential to prove the reliability and novelty of the software. As advised, we have now included experimental validation. We designed distinct structures, including (i) a higher-order RNA assembly formed from self-assembling RNA origamis, (ii) the largest RNA origami to date and (iii) an RNA droplet, showcasing the utility of pyFuRNace beyond classical RNA origami. The structures and their high-yield assembly was validated with AFM and confocal microscopy. The remaining comments are addressed in the responses below.

“For this manuscript, several issues require further clarification:”

Comment 1: “Experimental verification must be included: This is the most critical point for whether this paper can be accepted. The author should use PyFuRNace to design one or more RNA origami with novel topology or function from scratch. The design should highlight the unique advantages of PyFuRNace, such as taking advantage of its custom motif function or multi-chain design capability. Subsequently, the author needs to: Synthesize the RNA origami by *in vitro* transcription. Preliminary verification of its folding correctness and yield by gel electrophoresis (Native PAGE). Image the morphology of the nanostructure by atomic force microscopy (AFM) or transmission electron microscopy (TEM) and compare it directly with the 3D model generated by PyFuRNace. If the design contains functional elements (such as aptamers), corresponding functional experiments (such as binding experiments) should be performed to verify its activity. Only by providing these experimental data can the core argument of the paper - that PyFuRNace is a reliable and accelerated engine for RNA origami development - be convincing.

We appreciate the reviewer’s detailed suggestions, which closely guided our experimental validation pipeline. We tested pyFuRNace on three distinct RNA nanostructures:

1. **RNA Filaments:** Designed using pyFuRNace’s ”live 3D view” and ”multi-chain” features, optimising stem lengths for multimeric assembly.
2. **Large RNA Origami:** Designed using pyFuRNace’s default origami generation. To our knowledge, it’s the largest single-stranded RNA origami to date. It is composed of 2501 nucleotides, incorporates a Broccoli aptamer to demonstrate functional addressability and folds co-transcriptionally at high yield. Note that the previous largest RNA origami (Geary et al, Nat. Chem. 2021) was made of 2360 nucleotides, but the folding yield was extremely low (No structures without defects observed), while our large origami has an approximate folding yield of 80
3. **RNA Droplets:** Adapted the design of Stewart *et. al.* 2024 using pyFuRNace to generate sequence variants via its 5' start motif interface, preserving original kissing loops and a Broccoli aptamer. This demonstrates the utility of pyFuRNace beyond classical RNA origami.

Designs 1 and 2 were PCR-amplified using pyFuRNace-suggested primers and transcribed *in vitro*. Folding was verified via agarose gel electrophoresis (included in the revised Supplementary Information, Figure S6), and RNA nanostructures were imaged using AFM (presented in Figure 5 of the revised manuscript, as well as Figures S7 and S8). The AFM images confirmed that the filament morphology matched the predicted 3D model from pyFuRNace. The Broccoli aptamer functionality was confirmed by fluorescence in plate reader assays, which also reveals the rate of co-transcriptional folding (included in the revised Supplementary Information, Figure S6). Droplet formation and Broccoli activity were validated using confocal microscopy (presented in Figure 5 of the revised manuscript). All DNA templates and primer sequences are presented in

the revised version. These experiments confirm that pyFuRNace enables both reliable design and functional expression of diverse RNA architectures.

2 Claims on reliability should be more rigorous: The authors should adjust their wording without experimental data to support it. PyFuRNace undoubtedly improves the efficiency and convenience of design, but it is premature to claim that it improves reliability. The authors should clearly point out that the reliability of the tool currently relies on the intrinsic performance of its integrated third-party algorithms (such as Revolvr and ViennaRNA), while PyFuRNace assists designers to make more reliable decisions by optimizing the folding path checker and providing real-time visual feedback.

We thank the reviewer for highlighting the need for more rigorous wording regarding PyFuRNace's reliability. We believe that the new experimental validation helps confirm the reliability. We have revised the manuscript nevertheless to clarify that the tool's reliability depends in part on its integration with established third-party algorithms such as ViennaRNA and Revolvr. PyFuRNace assists users in making more reliable design decisions by offering folding path visualisation, real-time 3D feedback, and especially default parameters (e.g., wobble frequency, dovetail spacing) that are developed by prior experimental optimisation (Geary *et al.* 2021, Tran *et al.* 2024). Additionally, our experimental validation of multiple nanostructures—particularly the successful folding of the large RNA origami using default parameters—provides preliminary evidence that these defaults can help guide users toward reliable designs. We have added a Supplementary Note (Note S3) with the limitations of the dependencies, suggesting workaround solutions.

3. Add detailed comparison with other tools: Although the paper mentions tools such as ROAD, Tiamat, and Assemble2, the comparison is too brief. It is recommended to make a detailed functional comparison between PyFuRNace and existing key tools in the form of a table, from multiple dimensions such as supported design paradigms (co-transcription/TectoRNA, etc.), user interface (GUI/command line), whether it supports multi-chain, 3D visualization, custom motifs, script programming, simulation integration, etc. In particular, if PyFuRNace has advantages in specific performance or functions that other design methods cannot achieve at all.

We thank the reviewer for the valuable suggestion. In response, we have added a detailed comparison table and accompanying description in the Supporting Information (Note S1) and referred to it in the revised version of the main text. The table compares PyFuRNace with tools such as ROAD, Tiamat, and Assemble2 across multiple dimensions. We also added an extended description and discussion of the tools listed in the table.

We acknowledge that the usability and effectiveness of each tool can vary depending on the user's familiarity, expertise, and specific application. Therefore, we present the comparison as a general guideline rather than a definitive evaluation. Nonetheless, we believe the table highlights PyFuRNace's unique strengths that distinguish it from existing tools and helps to guide the

Response Figure 1: New Figure 5 showing experimental validation of pyFuRNace. Three distinct RNA nanostructures were designed and tested: **a)** A two-layer RNA origami tile that polymerizes into filaments. AFM (middle and right image) validates successful co-transcriptional folding according to the designed 3D model (left). **b)** A large rectangular RNA origami structure consisting of 2,501 nucleotides with a Broccoli aptamer (highlighted in green). AFM images confirm high-yield co-transcriptional folding. **c)** ARNA droplet-forming motif with kissing loops shown in blue and the Broccoli aptamer in green. A bright-field image (middle) and a confocal fluorescence image (right) confirm droplet formation.

reader within the landscape of existing tools.

a) RNA gel

b) Plate reader assay

Response Figure 2: New Figure S6 showing gels of the above mentioned RNA nanostructures and fluorescence measurements of the broccoli aptamer on the large 10-helix RNA origami

Response Figure 3: New Figure S9 showing the raw folding yield estimation for the large RNA origami design.

4. Provide a more thorough discussion of the limitations of the dependent tools: The authors acknowledge some of the limitations of the Revolvr sequence generation engine in the Discussion section, such as the lack of fine control over multi-strand optimization and kissing loop energies. This is worth acknowledging. To provide readers with a more comprehensive understanding, it is recommended that the authors explain earlier and more clearly in the main text how these limitations directly affect the design capabilities of the current version of PyFuRNace, and

a) 5' start in the arm

b) 5' start in the core

Response Figure 4: New Figure S10 showing design variants of the droplets. Comparison of droplets formation inserting the 5' start in the arm (a) or in the core (b) of the nanostar. The start position (and hence the position of the nick) does not seem to influence droplet formation significantly.

explain how users should circumvent these issues during the design process.

Following the reviewer's comment, we have added an extended description on how the limitations of underlying tools such as Revolvr affect the current design capabilities of pyFuRNace, particularly in areas like multi-strand sequence optimisation and the fine-tuning of kissing loop energies (Note S3). We also suggest practical strategies to help users mitigate these challenges during the design process. While we agree that transparency is essential for informed use, we have been careful not to overemphasise these limitations in the main text, out of respect for the significant contributions of existing tools. Nevertheless, we have deeply detailed the main limitations in the revised Supplementary Information. To further support the users, we are expanding the documentation and tutorials with specific guidance on how to work around these constraints. In parallel, we are actively developing new optimisation algorithms to address the issues with Revolvr, which we plan to publish in the future and incorporate into future versions of pyFuRNace.

5. Provide a more thorough discussion of the limitations of the dependent tools: The authors acknowledge some of the limitations of the Revolvr sequence generation engine in the Discussion section, such as the lack of fine control over multi-strand optimization and kissing loop energies. This is worth acknowledging. To provide readers with a more comprehensive understanding, it is recommended that the authors explain earlier and more clearly in the main text how these limitations directly affect the design capabilities of the current version of PyFuRNace, and explain how users should circumvent these issues during the design process.

This comment appears to be a duplicate of Comment 4, so we have addressed it in our response above, clarifying the dependency limitations and providing strategies to mitigate these issues.

In summary, PyFuRNace demonstrates strong potential to become a widely adopted and impactful tool in the field of RNA origami, facilitating the design process and accelerating progress in RNA nanotechnology. However, the current manuscript lacks experimental validation of the reliability and structural fidelity of the designs generated by the software. Moreover, the innovation and performance claims made in the paper remain unsubstantiated and are likely to be met with skepticism in the absence of empirical evidence. Therefore, in its present form, I do not believe the manuscript meets the publication standards of Nature Communications. That said, if the authors can provide experimental demonstrations showing that PyFuRNace enables the successful design and realization of RNA origami structures that are difficult or infeasible to construct using existing tools—such as ROAD—and further establish its general applicability across diverse design challenges, the work would have the potential to make a substantive contribution to the field. Such evidence would significantly strengthen the case for publication in a high-impact journal.

We appreciate the reviewer's thoughtful feedback, which we found both constructive and highly relevant. In response, we have thoroughly addressed all the issues raised, with particular focus on experimental validation to support the innovation and performance claims of PyFuRNace. Specifically, we have now included experimental demonstrations of three distinct RNA nanostructures designed using PyFuRNace: (1) RNA filaments designed via multi-chain assembly; (2) the largest single-stranded cotranscriptional RNA origami to date, incorporating a functional Broccoli aptamer; and (3) RNA droplets based on the design by Stewart *et. al.* 2024, with new sequence variants generated through PyFuRNace's motif tools. These structures were validated through in vitro transcription, gel electrophoresis, atomic force microscopy (AFM), and functional assays (plate reader and confocal microscopy). These results provide empirical support for the structural fidelity and practical utility of PyFuRNace across diverse design challenges. We have also revised the manuscript to clarify the scope and dependencies of PyFuRNace's reliability, added a detailed comparison with existing tools, and provided guidance on how users can work around current limitations. We believe these additions directly address the reviewer's

concerns and substantially strengthen the manuscript.

Reviewer #2

The article presents an integrated workbench-type design tool pyFuRNAce for supporting the design of RNA nanostructures in the "RNA origami" paradigm introduced by C. Geary, P. Rothmund and E. Andersen in Science 2014 and successfully pursued in a followup paper in Nature Chemistry 2021. The pyFuRNAce software integrates a textfile-based core toolkit ROAD (Nat Chem 2021), developed mostly by C. Geary, together with several visualization and analysis tools into a convenient and easy to use design platform with a nice GUI, a clear workflow, and no need to move and convert intermediate files between a mishmash of disjointed special-purpose tools.

*In a broad classification of DNA/RNA nanotechnology design tools, pyFuRNAce is *motif-based*, that is, the target structures are developed bottom-up by combining elementary design motifs (helices, crossovers, pseudoknots, aptamers,...) in a well-defined manner. (PyFuRNAce also provides the useful possibility of extending the family of design motifs according to the user's needs.) In this pyFuRNAce falls between strand-based tools, such as caDNAno and ENSnano, and top-down model-driven tools such as vHelix and DAEDALUS. (All of the latter of course being DNA, not RNA design tools.) The tradeoff in going from strand-based to motif-based to model-based tools is that as the convenience of use increases, the level of detailed control decreases. In the particular case of pyFuRNAce, while it is convenient and effective for pursuing designs in the RNA origami framework, it is also very specific to this particular way of doing things.*

Thus, for people working in the motif-based RNA origami framework, which seems to be growing into the mainstream approach to RNA nanostructure design, pyFuRNAce will be a welcome work-saver; but outside of this framework it will naturally be less relevant.

In addition to this general assessment, I have a number of comments and questions to the authors:

We thank the reviewer for taking the time to review our manuscript in detail and thank him/her for the generally positive feedback. We do believe that pyFuRNAce is broadly applicable beyond "standard" motif-based RNA origami and demonstrate that in the revised version of the manuscript by realizing a RNA droplet (see revised Figure 5). While RNA nanostructure design (motif-based or other forms) may be niche at the moment, part of the reason is the difficult entry into the field due to complex design pipelines. This is exactly what we want to address with pyFuRNAce – make RNA nanotechnology accessible to a broad community with broad and new applications. We hope to convince the reviewer of the broader utility of pyFuRNAce with the new data gathered throughout the revision process. We address the comments in detail below.

Abstract: *You might want to make it more clear that pyFuRNAce is specifically targeted to support RNA origami types of designs. (Cf. "pyFuRNAce [...] enables the design of complex*

RNA nanostructures”, “pyFuRNace reduces the entry barrier for RNA nanotechnology”.)

Following the reviewer’s suggestion, we have clarified in the abstract that pyFuRNace is specifically tailored for RNA origami, while retaining the broader term “RNA nanostructures” to reflect its applicability to other classes, such as RNA droplets, which we validated experimentally during the revisions.

Introduction; P. 2, l. 6: Provide some background to the statement “Scientists, naturally, asked whether we would be able to design RNA sequences with defined structures and functions.” In general or for some specific purpose? Citations?

We clarified the sentence by giving examples of how this question led to the development of SELEX and RNA nanostructure design. We also added relevant citations to support both areas.

Introduction; P. 2, l. -6 ff.: “Three RNA design paradigms: RNA architectonics, co-transcriptional RNA origami, paranemic cohesion RNA origami”. Would also annealed RNA origami be relevant? What about hybrid RNA-DNA designs, which seem to be a growing trend, perhaps even more so than pure RNA origami?

In the revised manuscript, we now note that thermal annealing is possible for RNA origami, even if less explored due to the advantage of co-transcriptional folding for *in vivo* transcription or synthetic biology. We acknowledge hybrid RNA-DNA nanostructures as an emerging complementary design space, while clarifying that they are beyond the scope of this work.

Introduction; P. 3, l. 4: “DNA origami design... arbitrary(?) structures...” -¿ “very general structures”

We appreciate the reviewer’s comment. During the revision of the manuscript, we revised the sentence and removed the word “arbitrary” to improve clarity and accuracy.

Introduction; P. 3, ll. 4-5: “DNA origami... Although the names are similar, co-transcriptional RNA origami has some unique features”. What is the actual structural or functional similarity underlying the similar names?

We modified the text following the reviewer’s doubts. Basically, both, RNA and DNA origami are methods to create addressable nanostructures with a raster-filled pattern, enabling precise spatial control for nanoscale assembly and functionalization. In both cases nucleic acids are folded, hence origami. One could argue that by the same logic protein design could also be called protein origami and we feel that the comparison with proteins does more justice to the complexity of RNA folds (enabled by the extra possibilities for hydrogen bonding due to the presence of the 2’OH). For precisely this reason, we prefer the term RNA nanostructure.

*Introduction; P. 3, ll. 14-15: "The limited development of RNA origami can largely be attributed to one key factor: the lack of user-friendly design tools." Is this really *the* limiting factor? What about complexity of experiments or challenging/costly materials?*

We admit that attributing the limited development of an entire field to a single factor is a strong claim. We have revised the manuscript to clarify this point. As of today, the cost of RNA origami materials (DNA fragments and transcription kits) and experimental techniques (AFM, TEM, Cryo-EM) is comparable to that of DNA origami. In contrast, the availability of intuitive, RNA-specific design tools has lagged behind considerably. For instance, a dedicated design platform for RNA origami, ROAD, only became available in 2021 and is still missing a fully user-friendly interface. On the flip side, RNA-based medicines are experiencing a tremendous rise, which makes RNA nanotechnology an interesting future field.

Introduction; P. 3, ll. 16-17: In parallel to this list of tools, in particular when listing Athena from the Bathe group, it would be appropriate to mention also vHelix/BSCOR from the Högberg group, which is fundamentally similar but earlier, and better known & more used than several of the others.

We have included and referenced vHelix/BSCOR in the revised manuscript.

Introduction; P. 3, ll. 25: "As far as RNA design software is concerned, the landscape is relatively sparse. Tiamat... Assemble2..." What about pyDAEDALUS from the Bathe group or DNAForge from the Orponen group?

We have now included and referenced both pyDAEDALUS and DNAForge in the revised manuscript and in the discussion of tools which was added to the SI.

Materials and methods: since pyFuRNace integrates so many existing tools (ROAD most significantly, but also ViennaRNA, Forna, oxView,...), how much new code has actually been contributed by the pyFuRNace team? Also, all the other integrated tools are multipurpose, but ROAD seems to be really the engine at the core of pyFuRNace: is it now appropriately credited?

As of the current revision, the pyFuRNace version 0.0.9 consists of approximately 10100 lines of original Python code (excluding comments and blank lines). This codebase is actively maintained, and the line count is evolving due to ongoing feature development and bug fixes. ROAD and other tools that pyFuRNace builds on are appropriately credited in the manuscript as well as on the web platform pyfurnace.de

While pyFuRNace integrates several external tools to support visualisation, structure prediction, and other tasks (e.g., ViennaRNA, Forna, and oxView), its core logic, class structure, data processing, and user interface have been developed entirely by our team.

Regarding the ROAD package, only the Revolvr sequence optimisation algorithm has been integrated directly into pyFuRNace. Revolvr is indeed central to the inverse folding step, generating RNA sequences from target structures. Importantly, pyFuRNace is designed with modularity in mind and supports the integration of alternative inverse folding tools. We chose Revolvr because it is experimentally validated and specifically optimised for RNA origami applications.

In response to the reviewer's suggestion, we have included both the current Python line count and a more explicit acknowledgement of Revolvr's role in the revised manuscript.

Materials and methods: it would be nice to have a clear listing or diagram of which external libraries have been integrated, to which parts of pyFuRNace, in which way (source, binary, API,...), and what the reuse licenses of each of these libraries are. I was looking for something like this in the SI but couldn't find the information.

We added the diagram of the dependencies, external libraries and licences to the revised SI of the manuscript (see below, Figures S1, S2 and Note S3). The diagram is referenced in the main text.

Results/Generate, Convert, and Prepare: the article contains now no experimental validation of the designs created with pyFuRNace. Would such validation be needed, and if not then please present an argument in the paper why the designs should be trusted, and/or if there are some particular cases where caution should be exercised.

In the revised manuscript, we added experimental validation of different RNA nanostructures. We included RNA droplets to test the versatility of the software towards nanostructure classes beyond classical RNA origami (see response to Reviewer 1, Figure 5 and several new SI Figures and Tables).

Results/Generate, Convert, and Prepare; P. 11, ll. 16-17: "It is important to note that Revolvr [...] does not yet support kissing loop-specific binding energies." Please explain what the issue is here, and provide some pointer(s) to the literature where this issue is discussed.

Revolvr currently applies a general energy range to all kissing loops, but does not allow users to assign distinct binding energies to individual kissing loop pairs. This limitation constrains the ability to tune specific interactions within complex RNA origami structures. In DNA origami, overhangs and staples enable control of intra- and inter-tile interactions. In cotranscriptional ss-RNA origami, these roles are mainly fulfilled by internal and external kissing loops. Without the ability to set distinct binding energies, it is difficult to modulate the strength of these interactions independently (e.g., to enforce strong internal and weak external binding). A workaround is possible: users can manually assign loop sequences with the desired energies. However, this reduces sequence flexibility and can negatively affect folding robustness and automation. Following the

suggestions of Reviewers 1 and 2, we have clarified this limitation in the Supplementary Note S3, its implications for pyFuRNace's design capabilities, and potential strategies to mitigate it. To our knowledge, this issue has not yet been directly addressed in the RNA origami literature, likely because most cotranscriptional systems to date have involved relatively small, simple assemblies. As RNA origami scales in size and complexity, we expect that fine-grained control over kissing loop energetics will become increasingly relevant and deserve dedicated investigation.

Programming versatile RNA origami; P. 12, ll. -6 ff.: Is this the script in SI Note S3? If so, then please insert here a pointer to it, and maybe some additional commentary in the Note itself.

The lines in question refer to the `simple_origami` function within the pyFuRNace API, which is also accessible through the user interface as a popover menu, so it does not refer to the script in SI Notes (previously S3). However, we acknowledge the need for clarification. Since Figure 5 has been replaced with the experimental validation of designs generated using this function, we now provide new SI notes containing the corresponding code (with commentary) and direct references to them. This addition supports the function's utility and provides the requested context and reproducibility.

Programming versatile RNA origami; P. 13, ll. 3-4: "If a pseudoknots pairs before a stem is completely formed, threading of the strand through the duplex is kinetically unfavored and thus is unlikely to form co-transcriptionally." Please explain more clearly what the issue is here, and provide some pointer(s) to the literature where this issue is discussed. Is the proposed optimization routine guaranteed to resolve this issue?

In the main text, we included the reference to the original work where the folding barrier is defined. The issue arises from the cotranscriptional nature of RNA origami folding, as described in the original ROAD paper. During transcription, the 5' end of the RNA begins to fold while the 3' end is still being synthesised and remains complexed with RNA polymerase and DNA template. This creates a directional and spatial constraint on folding pathways. The early formation of tertiary contacts locks the structure in a 3D topology that may sterically hinder the threading needed for subsequent helix formation. The RNA transcription complex may lack sufficient space or flexibility to allow a helix to wind correctly, leading to folding failure. The folding barrier metric used here is a simplification. In reality, cotranscriptional folding is influenced by multiple factors, including the presence of unpaired regions, the length and positioning of dovetails, and the overall RNA structure. While our optimisation routine helps mitigate these kinetic issues, by delaying the formation of tertiary contacts, it cannot guarantee correct cotranscriptional folding. In practice, the overall design of the nanostructure remains a critical element.

Discussion; P. 14, ll. 4-5: "[T]he lack of integrated and user-friendly design tools has hindered the broader adoption of RNA origami." What about e.g. challenges in the laboratory processes?

We addressed this comment in the response to a question above (see *Introduction; P. 3, ll. 14-15*).

Discussion; P. 15, ll. 11-13: "Additionally, the sequence generation engine (Revolvr) lacks support for multi-strand optimization and fine-grained control over individual kissing-loop energies. To address these issues, we are developing a new inverse folding algorithm [...]" Cf. comment to p. 11, ll. 16-17: Please explain what is/are the actual technical issues here, and how are they addressed now that the new inverse folding algorithm is not yet available?

We have expanded the discussion of this in the manuscript's limitations section (Supplementary Note S3). The core technical issue lies in the use of the RNAFold and Revolvr algorithms, which are primarily designed for single-stranded RNA systems. While ViennaRNA's CoFold can predict the folding of two interacting strands, Revolvr only supports inverse folding for individual RNA strands. A workaround for optimising multimeric designs consists of generating each tile sequentially.

References: The capitalization of article and journal titles is quite inconsistent across the reference list.

We addressed the issue in the reference list. We leave it up to the professionals in the production process to address any inconsistencies in capitalisation that we may have missed.

References; Ref. [29]: The bibliographic information here is incomplete.

Thanks for noticing! We completed the bibliographic information.

Reviewer #2 (Remarks on code availability): I conducted a cursory review of the code, which seemed very cleanly and professionally done; but I did not look too much into the details. In addition to testing the server-hosted version I also performed a local installation on my Mac following the instructions given in the documentation. Everything went smoothly except for me needing to upgrade my Python version to 10.3 as required by the tool. – As noted in my review, one thing I felt was missing in the documentation was a clear map of the dependencies on external libraries (incl. current version requirement, availability, license). A small general concern with all tools relying heavily on external libraries is that they may and will break down eventually when the libraries are updated, so require committed maintenance.

We thank the reviewer for taking the time to test both the server-hosted and locally installed versions of pyFuRNace, and we appreciate the positive remarks regarding code quality and installation instructions. We acknowledge the reviewer's suggestion regarding a clearer overview of external dependencies, including version requirements and licensing information. In response,

we have now added a dedicated section to the documentation that:

- Lists all major external dependencies used by pyFuRNAce (e.g., ViennaRNA, Forna, oxView, ROAD),
- Specifies the required version ranges,
- Indicates license types
- Provides installation and compatibility notes where relevant.

We agree that long-term tool sustainability is crucial, especially given the reliance on multiple external libraries. Moreover, the modular design of pyFuRNAce helps isolate and manage external interfaces, reducing the risk of cascading failures due to upstream changes. We are committed to maintaining the codebase and adapting to future changes in the software ecosystem, and we appreciate the reviewer's emphasis on this important aspect. We envision a similar model as is currently employed for other widely used tools such as OxDNA.

Reviewer #3

The manuscript by Morani et al. entitled "PyFuRNAce: An integrated design engine for RNA origami" describes a new user-friendly software platform that facilitates the design of RNA origami nanostructures. Apart from making the design workflow for RNA origami more easily accessible, the most noteworthy new features are the ability to design multi-stranded structures, a live 3D view, a large and extendable motif library, and the ability to automate design by scripting. The work is significant for the field since it (compared to earlier published scripts for RNA origami design) lowers the barrier for researchers to make their own advanced designs. The main result of the manuscript is a well-functioning platform and the design examples stay within realm of already experimentally verified RNA origami structures and multimer assemblies. The PyFuRNAce software package is professionally done, extendable, and very well documented. The accessibility of the software would benefit from a few video tutorials that show how to design concrete examples e.g. the nanotube assembly and scripting of larger sheets and bundles. The work represents an important contribution and will be highly useful for researchers in several areas of RNA research, and I therefore highly recommend its publication.

Reviewer #3 (Remarks on code availability): *The code for PyFuRNAce is maintained on Github and is well documented. Looking through the code it is well annotated, which will allow other researchers to easily extend the code.*

We sincerely thank Reviewer #3 for the thoughtful and encouraging feedback on our manuscript. We appreciate the suggestion to include video tutorials to further improve accessibility. In response, we have already appended one short tutorial (Video S1) and are preparing a set

of brief, focused video tutorials that will walk users through the design features, accompanied by corresponding Python scripts. These will be made available via the GitHub repository and documentation website shortly. We are also grateful for the positive remarks regarding the codebase and its extensibility. We aimed to create a tool that not only streamlines design workflows but also invites collaboration and customisation from the broader community. Thank you again for your kind recommendation and support of our work.

Response Figure 5: New Figure S8 with additional AFM images of the large RNA origami rectangle Rect-10H-3X.

a) Design module UML-like diagram

b) Motif classes UML-like diagram

Response Figure 6: **a)** UML-like diagram representing the class structure of the Design sub-package of pyFuRNACE. The dependencies are shown in smaller boxes, with green indicating required dependencies and grey indicating optional dependencies. **b)** UML-like diagram representing the currently available Motif classes in pyFuRNACE. The RNAfold dependency is needed only when using a custom sequence in a Kissing Loop to calculate the respective energy.

a) Generate and Prepare modules dependencies

b) Graphical user interface dependencies

Response Figure 7: **a)** General diagram to represent the organisation and dependencies of the Generate and Prepare subpackages of pyFuRNace. **b)** General diagram to represent the organisation and dependencies of the pyFuRNace graphical user interface (GUI), in the App subpackage. The GUI Design page depends on the pyFuRNace Design subpackage; the GUI Generate page depends on the pyFuRNace Generate subpackage, while the GUI Convert and Prepare pages depend on the pyFuRNace Prepare subpackage.

Point-to-point response

Reviewer #1

“The authors have addressed all my questions. I recommend the manuscript to publish in NC.”

We thank the reviewer for their positive evaluation and for recommending the manuscript for publication.

Reviewer #2

“The authors have addressed all the main questions in my earlier review most commendably, and the manuscript has in particular been much strengthened by the now included experimental validation of novel RNA structures designed using the pyFuRNACE tool. Also the clarity, precision and coverage of the writing has been improved quite a bit in several places. I hope the authors still have the opportunity to fix the few remaining small typos in the main manuscript and in the captions of the SI figures, as well as check the list of references once more for correctness and formatting (e.g. ref. [65]).”

We thank the reviewer for their positive assessment of our revisions and for recognizing the improvements in clarity, precision, and coverage. We checked the manuscript and the SI to correct the typos, and reviewed the reference list to remove inconsistencies (including the journal of ref. [65]).

“One question that the authors could have addressed more explicitly in their revised version of the manuscript is the kinetic folding barrier mentioned on p. 13, ll. 3-4 of the original submission. This is discussed in the authors’ rebuttal letter, but the issue seems to have been omitted from the revised manuscript. (The rebuttal letter indicates that “in the main text, we included the reference to the original work where the folding barrier is defined”, but I couldn’t identify this reference. The terms ‘pseudoknot’ or ‘folding barrier’ also no longer appear in the revised manuscript.) While this issue does not seem to have been a concern in the experimental work now presented in the paper, maybe the authors could still add some notes on this potential concern in the Discussion section of the final publication?”

We thank the reviewer for this accurate feedback. In our initial revision, we had expanded the discussion of folding barriers, but this was later replaced with the new experimental validation. Considering that the folding barrier calculations (Geary et al., Nat. Chem. 2021) are an extreme simplification of folding kinetics, and the folding pathway was not a concern in our experiments, we were uncertain whether to include this discussion. However, following the reviewer’s suggestion, and in light of the community’s growing interest in cotranscriptional folding pathways (e.g.,

Orponen et al., DNA31 Proceedings), we have now added an extended note on this limitation in the Software Limitations section of the SI, and we reference this addition explicitly in the Discussion section of the main text.

“Other than that, I am quite favorably impressed by the quality of the article. I am convinced that pyFuRNace will become a widely used tool in designing RNA origami nanostructures and will contribute significantly to the advancement of the area of RNA nanotechnology. (Remarks on code availability) I reviewed the code and tested the tool in my review of the original submission, and was convinced of its engineering quality.”

We thank the reviewer for the encouraging comments. We also greatly appreciate the reviewer’s effort in testing the code. We confirm that the code remains openly available with documentation and version control, ensuring transparency, reproducibility, and accessibility for the community.